# Disinfection of Reusable Laryngoscopes: A Survey about the Clinical Practice in Spain

**DOI:** 10.3390/healthcare11081117

**Published:** 2023-04-13

**Authors:** Manuel Á. Gómez-Ríos, José Alfonso Sastre, Teresa López, Tomasz Gaszyński

**Affiliations:** 1Department of Anesthesiology and Perioperative Medicine, Complejo Hospitalario Universitario de A Coruña, 15006 A Coruña, Spain; 2Anesthesiology, Perioperative Medicine and Pain Management Research Group, 15006 A Coruña, Spain; 3Spanish Difficult Airway Group (GEVAD), 15006 A Coruña, Spain; 4Department of Anaesthesiology, Salamanca University Hospital, 37007 Salamanca, Spain; 5Department of Anesthesiology and Intensive Therapy, Medical University of Lodz, 90-419 Lodz, Poland

**Keywords:** anesthesiology, tracheal intubation, laryngoscopy, ross infection, healthcare-associated infections

## Abstract

Airway device-associated infections resulting from the cross-contamination of reusable laryngoscopes are one of the main causes of healthcare-associated infections. Laryngoscope blades are highly contaminated with various pathogens, including Gram-negative bacilli, which can cause prolonged hospitalization, high morbidity and mortality risks, the development of antibiotic-resistant microorganisms, and significant costs. Despite the Centers for Disease Control and Prevention and the American Society of Anesthesiologists’ recommendations, this national survey of 248 Spanish anesthesiologists showed that there is great variability in the processing of reusable laryngoscopes in Spain. Nearly a third of the respondents did not have an institutional disinfection protocol, and 45% of them did not know the disinfection procedure used. Good practices for the prevention and control of cross-contamination can be ensured through compliance with evidence-based guidelines, education of healthcare providers, and audits of clinical practices.

## 1. Introduction

Healthcare-associated infections (HCAIs) are infections that begin 48 h or more after hospitalization or within 30 days after receiving healthcare, and they constitute the second most frequent adverse event suffered by hospitalized patients [1]. Thus, 5–15% of these patients acquire HCAIs [1], which represents more than 4 million admitted patients and 7 billion EUR in direct costs per year in Europe [2]. HCAIs cause prolonged hospitalization, high morbidity and mortality risks, the development of antibiotic-resistant microorganisms, and high costs [2]. Therefore, the goal must be to avoid the risk of these adverse events between patients and healthcare workers.

Among HCAIs, airway device-associated infections caused by cross-contamination are one of the most critical and problematic types [3]. Studies have shown a high risk of pathogen transmission when using reusable blades and handles on laryngoscopes [4,5]. Ayatollahi et al. [6] found that laryngoscope blades had the highest level of contamination within the medical apparatus with Gram-negative bacilli. Other researchers found pathogens such as *P. aeruginosa* [5,7,8], *S. aureus* [9,10], *Serratia marcescens* [11,12], *Klebsiella* [8], *E. coli* [8], *Streptococcus pneumoniae* [8], and the multiresistant *Acinetobacter baumannii*. The researchers also found pathogens associated with neonatal death [5,7,11,12]. Bacteria are the main pathogens, but prions, fungi, and viruses, including Coronavirus-19, hepatitis A, B, and C, and the human immunodeficiency virus (HIV), can be potentially transmitted by cross-contamination. The infections caused by cross-contamination between patients through reusable laryngoscopes could cost more than 35,000 EUR per case, which equates to 25 EUR per intubation.

Laryngoscopes are considered semi-critical items as they come into contact with patients’ respiratory mucosa and require high-level disinfection using chemical disinfectants [13,14]. The Center for Disease Control and Prevention (CDC) and the American Society of Anesthesiologists recommend cleaning and high-level disinfection or sterilization of laryngoscopes with an FDA-approved agent for 10 min. Autoclaves are considered the gold standard for sterilization [15]. Ideally, both the laryngoscope blade and handle should be autoclaved. However, their batteries must be replaced, reassembled, and function-checked before use, exposing them to repeated handling and, thus, potential contamination after sterilization [16]. Regulatory bodies, such as the Joint Commission (USA), emphasized the importance of standardizing the reprocessing and storage of laryngoscope blades and handles to reduce the risk of infection. Guidelines for the decontamination of laryngoscopes exist in several countries [17]. However, there are no specific guidelines related to laryngoscopes in Spain. Good practices for the prevention and control of cross-contamination can be ensured through compliance with evidence-based guidelines, education of healthcare providers, and audits of clinical practices [8].

We hypothesized that there is great variability in the processing of reusable laryngoscopes in Spain. The objective was to determine the current national practices for the disinfection and storage of laryngoscopes.

## 2. Methods

The local ethics committee determined that this study did not require formal evaluation as it involved an anonymous survey of healthcare professionals.

The inclusion criteria for the survey were as follows: Spanish anesthesiologists who work both in the field of public and private health and are members of the Spanish Society of Anesthesiology, Resuscitation, and Pain Therapy (SEDAR). The SEDAR sent a survey link in a Google Forms format to its partners via email on 19 November 2021. The questionnaires completed within one week after that date were included in the analysis. The survey included the main aspects of the disinfection and storage of laryngoscope blades and handles. Participation was voluntary and anonymous. The survey is shown in Table 1.

## 3. Results

We received 248 completed surveys from all over the national territory, which was a response rate of 6.2%. Only 5 of all 50 regions in Spain were not represented among the respondents. The main results of the survey are shown in Figure 1 and detailed below.

A total of 90% of professionals use devices with a reusable blade and handle, and 89% of those who do not have disposable equipment stated that this is due to avoiding cost increases.

A total of 31.5% of the respondents do not have an institutional disinfection protocol.

A total of 45% of the responders do not know the disinfection procedure used, while the most common procedure was high-level cleaning and disinfection (38%).

A total of 87% of anesthesiologists do not know the specific detergent used, while 65% do not know how long the blade should remain submerged in the disinfection solution, and 17% said that it should stay submerged for less than 10 min.

A total of 50% of the centers do not subject the handle to cleaning and high-level disinfection, and 34% would do so only if the handle is contaminated with secretions.

A total of 16% of the centers store their reusable laryngoscopes in sealed containers.

Disinfected/sterilized laryngoscopes do not offer the same guarantees as the single-use ones for 70% of the responders, and 79% consider that the single-use laryngoscopes avoid more infections due to the absence of cross-contamination.

## 4. Discussion

The present survey shows that, in the absence of national guidelines, professionals rely on institutional decontamination practices without knowing them in many cases. This work, similar to recent reviews, suggests the possible ineffectiveness of the disinfection methods for reusable laryngoscopes in Spain.

HCAIs can be extremely severe and, in some cases, even life-threatening, especially for patients with weakened immune systems or those who are already ill. These infections can lead to various complications, such as sepsis, pneumonia, and surgical site infections, and may cause extended hospital stays, increased healthcare costs, and, in the worst cases, patient deaths. Therefore, it is of the utmost importance to prevent and control nosocomial infections, which requires a multi-faceted approach to ensure patient safety and the overall quality of healthcare.

Laryngoscopes are essential tools in medical practices used for tracheal intubation. They are classified as semi-critical items due to their contact with the oral and respiratory mucosa. Thus, laryngoscopes can become contaminated with microorganisms, such as bacteria, viruses, fungi, or prions. Therefore, they pose a high risk of microorganism transmission to both patients and operators. These devices should be cleaned and disinfected using chemical disinfectants for a minimum of 10–45 min, following an agreed-upon protocol. However, in many instances, the handle is not included in the cleaning process, or the devices are not exposed to the disinfectants for a sufficient length of time. Additionally, only 16% of the surveyed facilities store the devices in sealed packaging, which is recommended to prevent contamination.

To minimize the risk of cross-contamination, improve safety, and reduce costs associated with decontamination and nosocomial infections, experts and organizations, such as the Joint Commission, recommend transitioning to high-quality, affordable, single-use laryngoscopes. However, in Spain, only 10% of healthcare facilities are currently able to adopt this recommendation.

Other studies on this subject have produced comparable findings to those of our survey. Esler et al. [18] performed a similar survey on 289 Royal College tutors in Great Britain. They found out that there are no guidelines relating to laryngoscope care in 60% of the units, while in 12% of the units, “disposable” laryngoscopes are used, but one-quarter of these units re-use the laryngoscopes. Almost two-thirds of the units use McCoy laryngoscopes, and yet, less than one-tenth of these change their routine cleaning procedures for them. None of the units distinguish between fibrelight and electric bulb-type blades in their cleaning methods. Bucx et al. [19] performed a survey of the decontamination procedures of laryngoscopes in Dutch hospitals. The study revealed that there are substantial differences in the decontamination procedures in Dutch hospitals compared with the situation in the UK [18]. The decontamination standards employed through manual methods were found to be unsatisfactory in a significant 78% of the 139 hospitals assessed. In 22.9% of these hospitals, manual cleaning was deemed insufficient, while manual disinfection failed to meet the established standards of the APIC (Association of Professionals in Infection Control and Epidemiology), CDC (Centers for Disease Control), and ASA (American Society of Anesthesiology) in all of the hospitals evaluated. Decontamination by instrument-cleaning machines was a standard procedure in 30 (22%) hospitals. In three of these hospitals, the blades were subsequently sterilized. No guidelines were observed for the decontamination procedures in Dutch hospitals at this time. Chawla et al. [17] found a similar situation in laryngoscope decontamination practices in India, concluding that there is a need to develop definitive guidelines on this subject.

Healthcare providers are required to adhere to rigorous infection control measures to prevent HCAIs associated with laryngoscopes. These measures include thoroughly cleaning and disinfecting reusable laryngoscopes after each patient use, as well as using single-use, disposable laryngoscope blades whenever feasible. While using sterile, single-use equipment can prevent cross-contamination and reduce reprocessing costs, concerns arise about the cost and environmental impact of supplying and disposing of this equipment. In addition to these measures, proper hand hygiene must be maintained before and after using laryngoscopes to further reduce the risk of HCAI transmission.

Our survey has several limitations. We simplified the survey in order to avoid respondent fatigue and obtain a greater number of respondents. Despite this, the response rate was low. This could mean that incomplete information was obtained in some areas of the survey and that the national practice has not been faithfully reflected given the high rate of non-responders. This increases the risk of bias. However, this survey may be a useful approximation to a problem in clinical practices that must be resolved.

## 5. Conclusions

The prevention of cross-contamination could require changes in the processing of reusable laryngoscopes given the inconsistencies and variability detected in practices throughout the national territory. Cost-effectiveness studies of disposable devices could stimulate the standardization of their use. National or international guidelines are needed to standardize the processing of reusable laryngoscopes.

Healthcare providers must be vigilant in their efforts to prevent HCAIs related to laryngoscopes to ensure patient safety and prevent the spread of infections in healthcare settings.

## Figures and Tables

**Figure 1 healthcare-11-01117-f001:**
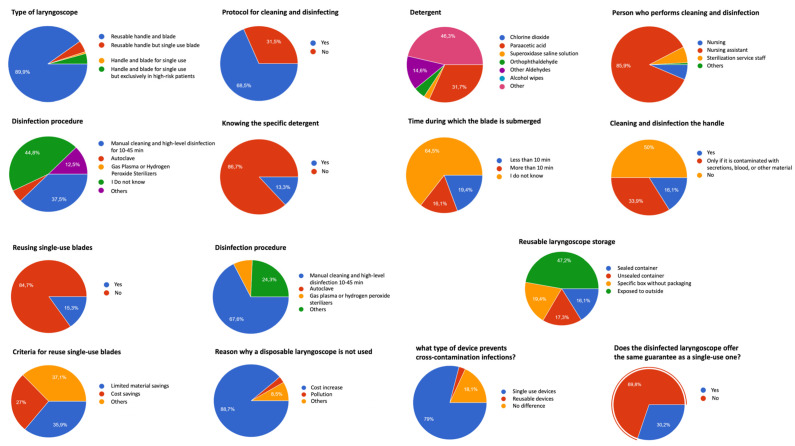
Main results of the survey.

**Table 1 healthcare-11-01117-t001:** Survey.

What type of laryngoscopes are utilized in your hospital? ○ Reusable handle and blade ○ Reusable handle with single use blade ○ Single-use handle and blade ○ Single-use handle and blade reserved exclusively for high-risk patients Does your hospital have a protocol in place for cleaning, disinfecting, or sterilizing reusable laryngoscopes? ○ Yes ○ No How long is the laryngoscope blade left submerged in the disinfection solution? ○ Less than 10 min ○ More than 10 min ○ I do not know Who is responsible for cleaning, disinfecting, or sterilizing the laryngoscope in your hospital? ○ Nursing staff ○ Nursing assistant ○ Sterilization service staff ○ Others: _________ What is the cleaning and disinfection or sterilization procedure for the laryngoscope? ○ Manual cleaning and high-level disinfection 10–45 min ○ Autoclave ○ Gas Plasma or Hydrogen Peroxide Sterilizers ○ Others: _________ ○ I Do not know Is the laryngoscope handle subjected to high-level cleaning and disinfection after every use? ○ Yes ○ Only if it is contaminated with secretions, blood, or other biological material ○ No Do you know the specific detergent used in your hospital for high-level disinfection? ○ Yes ○ No If yes, what is the detergent used in your hospital for high-level disinfection? ○ Chlorine dioxide ○ Paraacetic acid ○ Superoxidase saline solution ○ Orthophthaldehyde ○ Other Aldehydes ○ Alcohol wipes ○ Other__________ Based on your experience, does a disinfected/sterilized laryngoscope offer the same level of protection against cross-contamination as a single-use laryngoscope? ○ Yes ○ No Do you ever use single-use laryngoscope blades for multiple patients? ○ Yes ○ No If yes, what disinfection procedure is performed on the single-use blade? ○ Manual cleaning and high-level disinfection for 10–45 min ○ Autoclave ○ Gas Plasma or hydrogen peroxide sterilizers ○ Others: _________ What criteria do you use to determine when to use a single-use blade on multiple patients? ○ Limited material savings ○ Cost savings ○ Others __________ How is the disinfected/sterilized reusable laryngoscope stored? ○ Sealed container ○ Unsealed container ○ In a specific box without packaging ○ Exposed to the outside Why do you think your hospital does not use disposable laryngoscope sets? ○ Increased cost ○ Pollution ○ Others:__________ Based on your experience, which type of device do you believe is more effective in preventing cross-contamination infections? ○ Single-use devices ○ Reusable devices ○ No difference __________

## Data Availability

Not applicable.

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
