# Peer review of "Disinfection of Reusable Laryngoscopes: A Survey about the Clinical Practice in Spain"

_healthcare, 2023, doi:10.3390/healthcare11081117_

Round 1

Reviewer 1 Report

Thank you for addressing an interesting topic (often neglected by anesthesiologists).

Please see my comments in the manuscript attached.

Author Response

Our comments on your suggestions are included in the PDF itself.

Thank you for your time and valuable comments.

Reviewer 2 Report

The paper by Gomez-Rios et al. touches on an important issue i.e. cleaning of scopes, here laryngoscopes, in hospitals and specialist practices. In this case cleaning is fairly easy as compared to flexible gastroscopes and the like. As cited, it is a clear risk for a new patient to be examined with a contaminated laryngoscope and a protocol for cleaning (sterilization if possible) is mandatory in the modern hospital. The paper is well written and the data sound.

I have som suggestions for improving the manuscript:

1. I cannot find the data for how many surveys were sent/how many anesthesiologist were contacted ? There should be a percentage of the 289 responses in relation to total sent.

2. If possible, the data would benefit from stratification in major, minor hospitals or government/private.

3. Among possible bacteria that can pose a contamination problem i would also mention Mycobacterium spp. - i couldn´t find any reports about this for larungoscopes, but mycobacteria can be present in the oral flora, and they may be even more difficult to eradicate - autoclave optimal.

4. All bacterial names should be in italics.

5. The questionnaire needs some better English translation, e.g. "How long does the blade remain..."; "Nursing", better "Nurse - or nurses"; "Based on his experience..", based on "your" - it could be a female responding; "How is the disinfected....laryngoscope stores?". And text in Results: "did not obtain respresentation of respondents in only 5 regions..", better: "only 5 of all 50 regions in Spain were not represented among the respondents".

6. Explain SEDAR in Methods

Author Response

Thank you for your time and valuable comments.
